

# Small population size and possible extirpation of the threatened Malagasy poison frog *Mantella cowanii*

Devin Edmonds[1,2], Raphali Rodlis Andriantsimanarilafy[3], Angelica Crottini[4,5,6], Michael J. Dreslik[1,2], Jade Newton-Youens[7], Andoniana Ramahefason[8], Christian Joseph Randrianantoandro[8] and Franco Andreone[9]

[1] Department of Natural Resources and Environmental Sciences, University of Illinois at Urbana-Champaign, Champaign, IL, United States
[2] Illinois Natural History Survey, Prairie Research Institute, University of Illinois at Urbana-Champaign, Champaign, IL, United States
[3] Madagasikara Voakajy, Antananarivo, Madagascar
[4] Departamento de Biologia, Faculdade de Ciências, Universidade do Porto, Porto, Portugal
[5] CIBIO, Centro de Investigação em Biodiversidade e Recursos Genéticos, Vairão, Portugal
[6] BIOPOLIS Program in Genomics, Biodiversity and Land Planning, CIBIO, Campus de Vairão, Vairão, Portugal
[7] Department of Natural Sciences, The Manchester Metropolitan University, Manchester, United Kingdom
[8] Mention Zoologie et Biodiversité Animale, Université d'Antananarivo, Antananarivo, Madagascar
[9] Museo Regionale di Scienze Naturali, Torino, Italy

Corresponding author
Devin Edmonds, dae2@illinois.edu

## ABSTRACT

Amphibians are experiencing severe population declines, requiring targeted conservation action for the most threatened species and habitats. Unfortunately, we do not know the basic demographic traits of most species, which hinders population recovery efforts. We studied one of Madagascar's most threatened frog species, the harlequin mantella (*Mantella cowanii*), to confirm it is still present at historic localities and estimate annual survival and population sizes. We surveyed eleven of all thirteen known localities and were able to detect the species at eight. Using a naïve estimate of detection probability from sites with confirmed presence, we estimated 1.54 surveys (95% CI [1.10–2.37]) are needed to infer absence with 95% confidence, suggesting the three populations where we did not detect *M. cowanii* are now extirpated. However, we also report two new populations for the first time. Repeated annual surveys at three sites showed population sizes ranged from 13–137 adults over 3–8 years, with the most intensively surveyed site experiencing a >80% reduction in population size during 2015–2023. Annual adult survival was moderately high (0.529–0.618) and we recaptured five individuals in 2022 and one in 2023 first captured as adults in 2015, revealing the maximum lifespan of the species in nature can reach 9 years and beyond. Our results confirm *M. cowanii* is characterized by a slower life history pace than other *Mantella* species, putting it at greater extinction risk. Illegal collection for the international pet trade and continued habitat degradation are the main threats to the species. We recommend conservation efforts continue monitoring *M. cowanii* populations and reassess the International Union

for Conservation of Nature (IUCN) Red List status because the species may be Critically Endangered rather than Endangered based on population size and trends.

## INTRODUCTION

Amphibian species are facing extinction rates at least 22 times faster than the average rate during the 10 millennia before industrialization, resulting in their status as the most threatened vertebrate class (*Ceballos et al., 2015*; *Luedtke et al., 2023*). Many species are experiencing severe population declines, leading to widespread range contractions through extirpation (*e.g.*, *Beyer & Manica, 2020*; *Granados-Martínez et al., 2021*; *Patla & Peterson, 2022*). Habitat loss is the largest threat to amphibians, but infectious diseases, invasive species, climate change, overexploitation, and pollution are all responsible for declines and interact in complex ways (*Collins, 2010*; *Grant, Miller & Muths, 2020*). Such threats and population trends highlight the immediate need for increased conservation, especially targeted toward the most threatened species and their habitats.

The island of Madagascar supports extraordinary amphibian species richness and endemism, with more than 415 described endemic frog species representing five anuran clades of independent origin (*Crottini et al., 2012b*; *Antonelli et al., 2022*; *AmphibiaWeb, 2023*). Alarmingly, 46.4% of assessed Malagasy frog species are threatened, owing largely to deforestation (*Ralimanana et al., 2022*; *IUCN, 2023*). Deforestation has eliminated as much as a quarter of the tree cover on the island over the last 25 years and the rate has only increased since 2005 (*Vieilledent et al., 2018*; *Suzzi-Simmons, 2023*). Consequently, many frog species in Madagascar have patchy distributions restricted to isolated pockets of forest in an otherwise inhospitable landscape (*Lehtinen & Ramanamanjato, 2006*). So far, there have been no documented modern frog species extinctions in Madagascar (*Andreone et al., 2008*, *2021*), but many records of species presence are from biological inventories conducted decades ago in areas with high rates of land use change. Verifying species presence and confirming the extant distribution of threatened species is some of the most vital information for informing conservation (*Villero et al., 2017*). Relatedly, we know little about frog population trends in Madagascar, even for highly threatened species. The lack of demographic information is not unique to Madagascar; we do not know survival or fertility rates for 87.5% of amphibian species globally (*Conde et al., 2019*). Baseline estimates of survival, recruitment, and other demographic traits are urgently needed to improve conservation efforts and inform management decisions (*Grant, Miller & Muths, 2020*).

Some of the most well-known amphibians in Madagascar are the Malagasy poison frogs in the genus *Mantella*. One species (*M. laevigata*) exhibits parental care and all *Mantella* species display aposematic coloration to warn predators of their poisonous skin alkaloids sequestered from prey (*Vences et al., 2022*). As such, they are familiar examples of

convergent evolution with Neotropical dendrobatids (*Daly, Highet & Myers, 1984*; *Chiari et al., 2004*; *Fischer et al., 2019*). While several *Mantella* species are widespread and have been found in degraded habitat and agricultural plantations (*e.g.*, *M. betsileo*, *M. ebenaui*, and *M. viridis*, *Vences, Glaw & Bohme, 1999*; *Andreone, Mercurio & Mattioli, 2006*; *Crottini et al., 2012a*), most are restricted to small areas, have highly localized populations, show a recent dramatic demographic decline, and are threatened by ongoing habitat changes (*e.g.*, *Crottini et al., 2019*). Compounding the threat of habitat loss is overexploitation; thousands of wild poison frogs are exported annually from Madagascar for the international pet trade (*Rabemananjara et al., 2007b*), though export quotas have been restricted recently to smaller quantities of just six species (*CITES, 2022*).

The harlequin mantella frog (*M. cowanii*) is one of the most threatened *Mantella* species, with a small and fragmented distribution in the central highlands. This region of Madagascar was formerly a mosaic of grassland, woodland, and subhumid forest, covering the mountainous area between the island's humid east and dryer west (*Yoder et al., 2016*). Today the central highlands consist mostly of secondary grasses and land converted for subsistence agriculture and cattle grazing, with little humid forest remaining (*Andriambeloson et al., 2021*; *Ranarilalatiana et al., 2022*). Thirteen localities of *M. cowanii* are known from the region: six in a cluster around the village of Antoetra, five 80 km northwest on the Itremo Massif, and two isolated localities located >100 km north of all other known populations, one near Betafo and the other east of Antakasina (*Rabibisoa, 2008*; *Rabibisoa et al., 2009*). One of the populations near Antoetra consists mostly of the closely-related *M. baroni* and *M. baroni* × *M. cowanii* hybrids (*Chiari et al., 2005*). All populations occur along mountainous streams with large boulders and adjacent wet rockfaces, which are typically covered in wet moss and other bryophytes. While some sites have intact gallery forests, others are almost entirely devoid of trees (Fig. 1). Frogs are mainly active at dawn and dusk and are only detectable during the rainy season (*Andrianasolo, 2016*; *Newton-Youens, 2017*).

Crucially, some *M. cowanii* populations are known from only one or two scientific expeditions carried out in the early 2000s (*e.g.*, *Andreone & Randrianirina, 2003*; *Andreone et al., 2007*; *Crottini et al., 2011*) and could now be extirpated. Due to its striking black and orange coloration (Fig. 2), *M. cowanii* was heavily exploited for the international pet trade during the 1990s and early 2000s (*Andreone, Mercurio & Mattioli, 2006*). From 1994–2003, several thousand frogs recorded as *M. cowanii* were exported from Madagascar for commercial purposes, after which legal trade was halted and the export quota was set to zero (*Rabemananjara et al., 2007b*; *CITES, 2022*). Despite heavy collection pressure and ongoing habitat loss, the demographic characteristics of *M. cowanii* populations remain largely unknown.

In 2008, a conservation strategy for *M. cowanii* was spearheaded by the IUCN Amphibian Specialist Group of Madagascar and Conservation International, which focused on improving habitat management at two localities near Antoetra (*Rabibisoa, 2008*). A decade later, a workshop was organized to update, build on, and revitalize the initial conservation strategy. The 2018 workshop participants included officials from Malagasy government, academia, biodiversity conservation organizations, and local

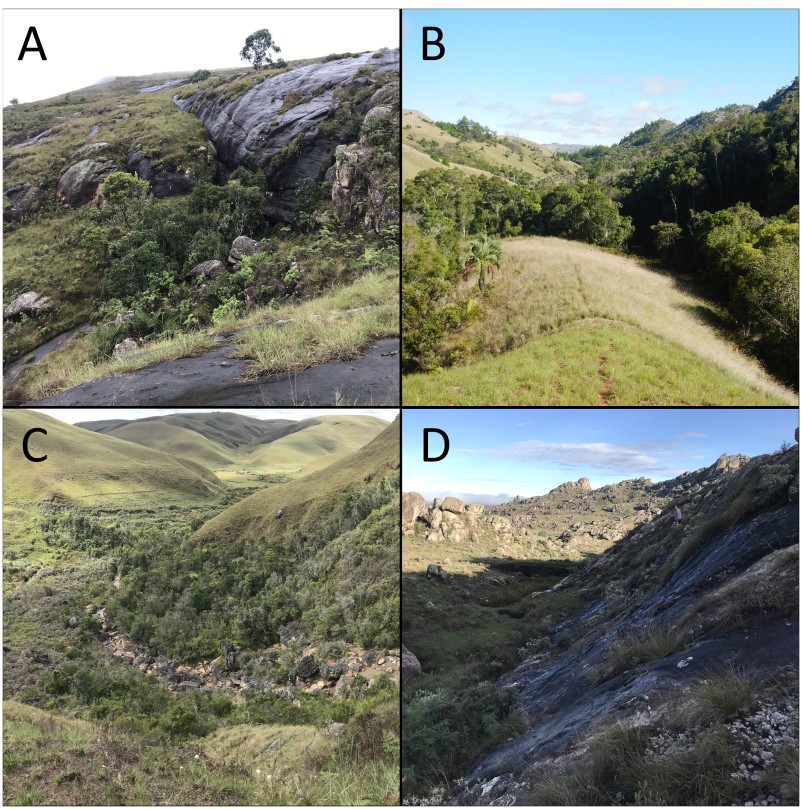

**Figure 1 Varying amounts of humid forest remaining at four *Mantella cowanii* localities in the central highlands of Madagascar.** The sites are (A) Fohisokina (B) Antsirakambiaty (C) Antakasina (D) Ambatofotsy. Only (B) is legally protected, falling within the boundaries of the Itremo Massif Protected Area.

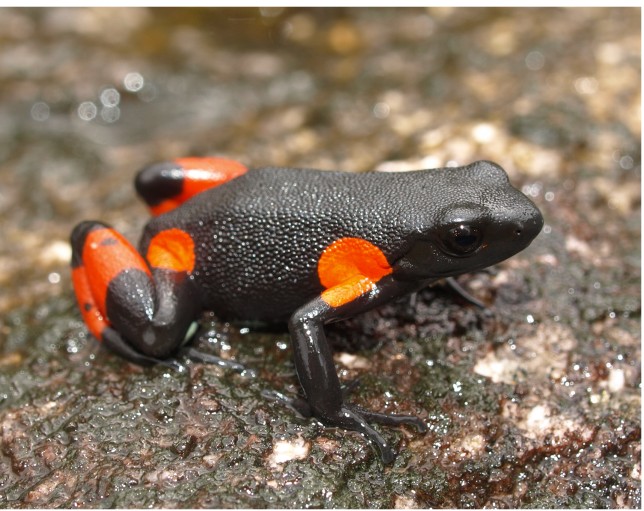

**Figure 2 The harlequin mantella frog (*Mantella cowanii*) at Soamasaka near Antoetra, Madagascar.** Its aposematic coloration led to high demand from the international pet trade. During the 1990s and early 2000s several thousand individuals were commercially exported from Madagascar. Since 2004, the commercial export quota from Madagascar has been maintained at zero.

communities (*Edmonds, Andreone & Crottini, 2022*). The workshop resulted in the *Mantella cowanii* Action Plan (McAP), which was officially launched in 2021 (*Andreone et al., 2020*; *Rakotoarison et al., 2022*). The McAP proposed 38 conservation actions needed for *M. cowanii*, with actions grouped into five themes: habitat protection, scientific research, local development, environmental awareness, and long-term sustainability. We aimed to fill the most critical research needs in the McAP by 1) confirming the presence of *M. cowanii* at localities across its range and 2) if present, estimating the key demographic traits of survival and population size.

## MATERIALS AND METHODS

### Study sites

We surveyed 11 of the 13 *M. cowanii* localities known from the literature and identified at the McAP workshop, nine in the Amoron'i Mania Region and two in the Vakinankaratra Region (Fig. 3). Sites ranged in elevation ~1,380–2,120 m asl. We repeatedly surveyed three sites (Ambatofotsy, Soamasaka, and Fohisokina) during 2020–2024 for 2–7 days/year to estimate demographic traits (Table 1). The sampling effort varied because of logistical constraints and security concerns, limiting our ability to survey for the same number of days annually. We combined these data from 2020–2024 with a survey at Fohisokina in 2015, which lasted 20 days to accomplish additional research objectives related to habitat use (*Newton-Youens, 2017*). The remaining eight sites (Ambinanitelo, Andraholoma, Antakasina, Antsirakambiaty, Bekaraka, Farihimazava, Tsimabeomby, and Vatolampy) were visited up to two times for 1–6 days to confirm species presence.

   We worked closely with local communities during fieldwork, surveying sites together with people from nearby villages. To identify additional sites, we asked if they had seen *M. cowanii* in other areas and encouraged people to watch for the species in new locations. Only two sites have some form of habitat protection: Fohisokina (also known as Vohisokina) is community-managed with an NGO (*Nowakowski & Angulo, 2015*), and Antsirakambiaty falls within the boundaries of Itremo Massif Protected Area (*Alvarado, Silva & Archibald, 2018*). The other sites are unmanaged and without legal protection.

### Data collection

We conducted fieldwork from late November to mid-January, surveying for frogs during 5–8 h. In 2015, we also surveyed Fohisokina during 16–18 h and combined these data with surveys from the morning of the same day. We did not survey during 16–18 h in 2020–2024 or at other sites due to safety concerns about traveling after dark. We searched for frogs visually, walking together in teams of 2–7 people along the stream or adjacent wet rockface. Our search extended opportunistically up to 10–15 m from the stream or wet rock wall. If we heard a frog calling, we used the call to help find its location. However, because the call of *M. cowanii* sounds nearly identical to that of *M. baroni*, and the two species hybridize at one site (*Chiari et al., 2005*), we did not rely solely on acoustic surveys. Stream segment length varied from 38 m at Bekaraka to 690 m at Ambatofotsy. At Fohisokina, we surveyed along six 50 m-long transects rather than opportunistically

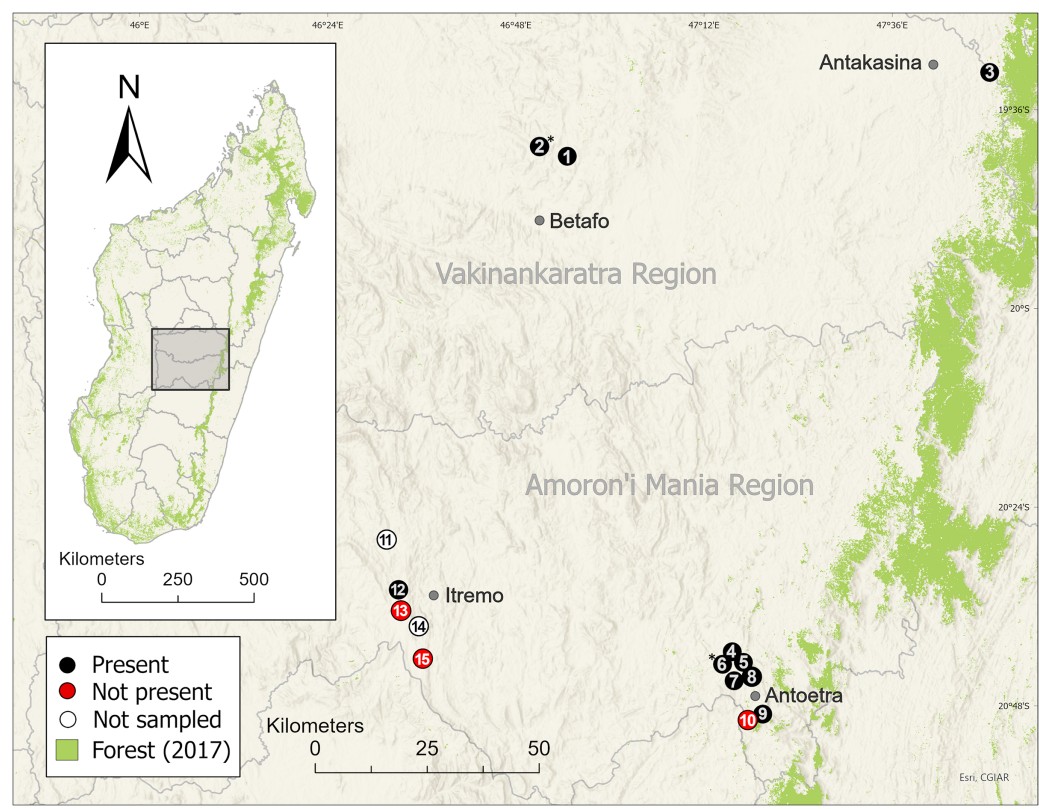

**Figure 3** **Distribution of the harlequin mantella frog (*Mantella cowanii*) in the central highlands of Madagascar.** Historically, the highlands were a forest-grassland mosaic but are now mostly secondary grassland used for grazing and subsistence agriculture. We sampled eleven of thirteen known *Mantella cowanii* localities, found frogs present at eight, and discovered two new localities. The gray lines are regional political boundaries. The green forest cover is from the 2017 update of *Vieilledent et al. (2018)*. The basemap layer was provided by ESRI. Betafo, Antakasina, Itremo, and Antoetra are towns (gray dots) used to refer to the four population clusters. The localities (and alternative name or spelling) are: 1 = Ambatofotsy, 2 = Sahandriana*, 3 = Antakasina (Antratrabe), 4 = Fohisokina (Vohisokina), 5 = Ambinanitelo, 6 = Ambohitsiholiholy*, 7 = Soamasaka (Soamazaka), 8 = Bekaraka, 9 = Farihimazava (Farimazava), 10 = Vatolampy (Maromanoa), 11 = Alan'i Volamena, 12 = Antsirakambiaty, 13 = Tsimabeomby, 14 = Andaobatofotsivava, 15 = Andraholoma. Those marked with * are new localities.

throughout the entire site to accomplish additional research objectives (see *Newton-Youens, 2017*).

After capturing a frog, we held it in a plastic bag (or a petri dish in 2015 at Fohisokina), marking the location with a GPS point, flagging tape, and a unique number. We measured snout-to-vent length to the nearest mm with digital calipers or a ruler. Individuals <22 mm were recorded as subadults under 1-year post-metamorphosis based on *Guarino et al. (2008)*. Weight was recorded with a digital scale to the nearest 0.01 g, and sex was recorded based on whether an individual had been calling before capture and, if not, body size (*Tessa et al., 2009*). We also took dorsal and ventral photographs, allowing us to identify recaptured individuals because each frog has a unique ventral pattern (Fig. 4). To a certain

**Table 1 Number of *Mantella cowanii* captured over five field seasons and survey effort.** Season refers to late November through mid-January of the following year. Days is the number of days a site was surveyed within a season. Captures is the total number of captures made. Individuals is the number of unique individuals caught within the season. An asterisk indicates the population consists of *M. baroni* × *M. cowanii* hybrids. Superscripts refer to the literature where the locality was first mentioned, with the name of the site in parentheses as it was written in the first report if different from the table below. Sites and localities correspond to Fig 3.

| Site | Locality | Season | Days | Captures | Individuals |
|---|---|---|---|---|---|
| **Antoetra** | Ambinanitelo[4] | 2022 | 2 | 14 | 13 |
| | Ambohitsiholiholy[7] | 2022 | 1 | 1 | 1 |
| | | 2023 | 2 | 18 | 18 |
| | Bekaraka[4] | 2021 | 3 | 3 | 3 |
| | | 2023 | 2 | 0 | 0 |
| | Farihimazava[2*] | 2022 | 4 | 71 | 63 |
| | | 2023 | 1 | 15 | 15 |
| | Fohisokina[1] | 2015 | 20 | 356 | 102 |
| | | 2020 | 5 | 69 | 34 |
| | | 2022 | 6 | 32 | 18 |
| | | 2023 | 6 | 24 | 11 |
| | Soamasaka[1] | 2020 | 5 | 31 | 19 |
| | | 2021 | 5 | 38 | 16 |
| | | 2022 | 5 | 35 | 18 |
| | | 2023 | 7 | 33 | 10 |
| | Vatolampy[2] | 2022 | 6 | 0 | 0 |
| | | 2023 | 2 | 0 | 0 |
| **Antakasina** | Antakasina[3] | 2021 | 3 | 1 | 1 |
| **Betafo** | Ambatofotsy[5] | 2021 | 2 | 16 | 12 |
| | | 2022 | 5 | 23 | 16 |
| | | 2023 | 7 | 35 | 22 |
| | Sahandriana[7] | 2023 | 3 | 21 | 15 |
| **Itremo** | Andraholoma[6] | 2023 | 2 | 0 | 0 |
| | Antsirakambiaty[4] | 2022 | 5 | 9 | 6 |
| | | 2023 | 5 | 5 | 5 |
| | Tsimabeomby[4] | 2022 | 1 | 0 | 0 |

Notes:
[1] *Ravoahangimalala et al. (2004)* (Soamantsaka; Vohitsokina).
[2] *Chiari et al. (2005)* (Farimazava).
[3] *Andreone et al. (2007)* (Antratrabe).
[4] *Rabibisoa (2008)*.
[5] *Rabibisoa et al. (2009)*.
[6] Unpublished inventory in the early 2000s.
[7] Present study.

extent, such photographic capture-mark-recapture techniques can be more accurate and are less invasive than traditional toe-clipping or visual implant elastomers (*Caorsi, Santos & Grant, 2012*; *Davis, VanCompernolle & Dickens, 2020*). Though software exists to help automate the photo-matching process (*sensu Edmonds, Kessler & Bolte, 2019*), we found

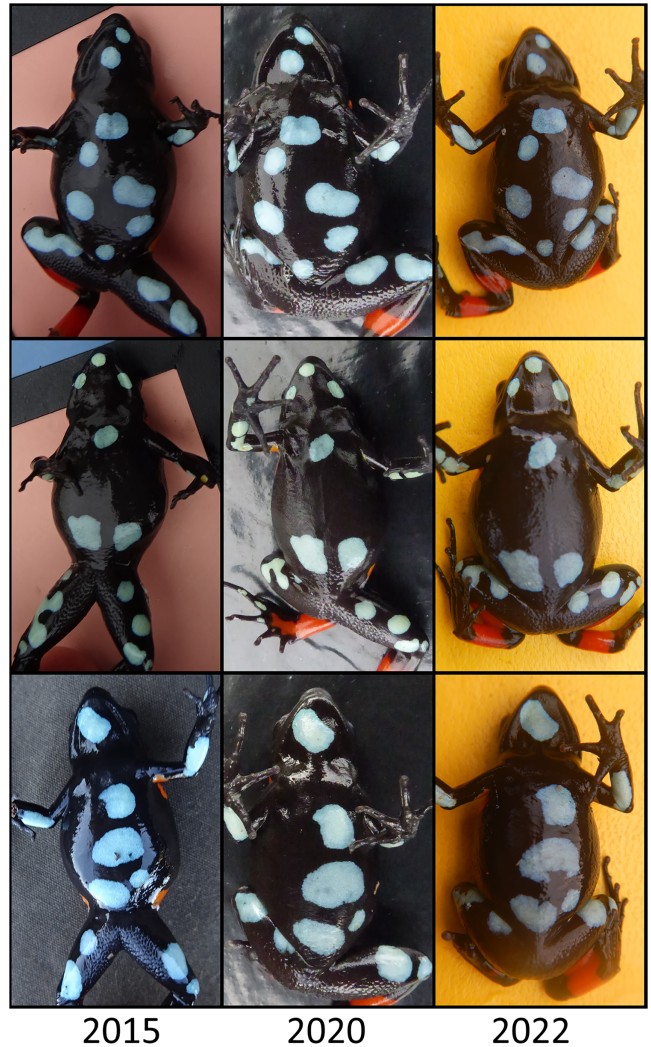

2015      2020      2022

**Figure 4** **Ventral photographs of *Mantella cowanii* from Fohisokina near Antoetra, Madagascar.** We used photographic capture-mark-recapture methods to estimate demographic traits. The ventral patterns of frogs allow for individual identification without marking. Six individuals originally captured in 2015 as adults were recaptured in 2022 and one in 2023. Three individuals captured in 2015, 2020, and 2022 are shown in the figure.

photograph angle and quality varied between days, sites, and years, so the automated process was prone to false negatives. Instead, one of us (D. Edmonds) visually examined all ventral photographs side-by-side and recorded when there was a match. Following photographs and measurements, frogs were released in the location where they were found, typically within 1–3 h after capture.

We followed all applicable international, national, and institutional guidelines for the care and use of animals. The study methods were approved by the Ministère de l'Environnement et du Développement Durable in permits N°173/20/MEDD/SG/ DGGGE/DAPRNE/SCBE.Re, N°439/21/MEDD/SG/DGGGE/DAPRNE/SCBE.Re, and

N°173/22/MEDD/SG/DGGGE/DAPRNE/SCBE.Re and by the University of Illinois Urbana-Champaign Institutional Animal Care and Use Committee in protocol #21180.

## Analysis

We used a robust design capture-mark-recapture model (*Pollock, 1982*) to estimate population size ($\hat{N}$) and apparent annual survival ($\varphi$) at Ambatofotsy, Fohisokina, and Soamasaka. These sites were selected because we had sampled them for at least 3 years, the minimum required for estimating annual survival when detection is imperfect. The closed robust design uses primary periods when the population is assumed open to births, deaths, immigration, and emigration to estimate $\varphi$ and secondary periods when the population is assumed closed to estimate $\hat{N}$. We used annual surveys from late November to mid-January as open primary periods and days as closed secondary periods (Table 1).

Our analysis compared 17 models, all incorporating site as a group-level effect. We considered models with either constant or site-specific $\varphi$ and the temporary emigration parameters $\gamma''$ and $\gamma'$ either constrained to 0 assuming no movement or set equal assuming random emigration. To account for individual heterogeneity in capture probability ($p$), we included a random effect of individual on $p$. Additionally, we compared models with capture probability covariates of site, year, number of surveyors, and survey effort calculated as the number of surveyors multiplied by the survey duration in minutes. We could not include environmental covariates that might be associated with capture probability because environmental variables were not collected consistently across all sites and years. To compare candidate models, we ranked them using Akaike's Information Criterion adjusted for small sample sizes (AIC$_c$; *Burnham & Anderson, 2002*). Models with $\Delta$ AIC$_c$ < 2 were considered to have support. We analyzed capture-mark-recapture data in program MARK through the *RMark* interface (*White & Burnham, 1999*; *Laake, 2013*) and assessed the goodness-of-fit with package *R2ucare* in R version 4.2.0 (*Gimenez et al., 2018*; *R Core Team, 2022*).

To infer absence if we did not detect *M. cowanii* at a historic locality, we estimated detection probability with a single-season occupancy model in package *unmarked* (*Fiske & Chandler, 2011*). Data from seven localities with confirmed presence and at least three surveys were used to generate a naïve estimate of detection probability assuming constant detection and occupancy. We then followed *Pellet & Schmidt (2005)* to estimate the number of surveys needed to detect the species as:

$$N_{min} = \frac{\ln(0.05)}{\ln(1 - p)}$$

where $N_{min}$ is the minimum number of surveys, $p$ is the detection probability, and 0.05 is the confidence level needed to be 95% certain of absence, assuming independent and comparable surveys. Using the number of surveys, we then calculated the confidence level around an observed absence given $N$ number of surveys as:

$$conf = e^{N*\ln(p)}$$
## RESULTS

### Verifying species presence

We confirmed *M. cowanii* presence at 8 of 11 surveyed localities and identified two previously unrecorded populations. However, we failed to detect the species at the historical localities of Andraholoma, Tsimabeomby, and Vatolampy. The naïve detection probability from sites with confirmed presence was 0.86 (95% CI [0.72–0.93]), showing it takes 1.54 surveys (95% CI [1.10–2.37]) to be 95% confident a population is extirpated and 2.37 surveys (95% CI [1.69–3.65]) to be 99% confident. With at least 2 days of surveys at Andraholoma and Vatolampy during suitable climatic conditions, we can be >97.9% (95% CI [91.9–99.9%]) confident the populations are extirpated and 85.7% (95% CI [71.7–93.4%]) confident there are no *M. cowanii* at Tsimabeomby.

### Capture patterns across sites

We made 764 captures of 280 individuals across all study sites and years, excluding putative *M. baroni* × *M. cowanii* hybrids. All but eight frogs were adults. Over half of all captures were at Fohisokina, where we caught 149 individuals 481 times. Six individuals at Fohisokina were recaptured 7 years after initially being caught in 2015 as adults (Fig. 4), one of which was recaptured again in year 8 in 2023. At our second most intensively surveyed site, Soamasaka, we made 137 captures of 40 individuals during annual fieldwork in 2020–2023. Three frogs were recaptured 4 years after their initial encounter in 2020, and 4 were recaptured 3 years apart. At Ambatofotsy, we conducted surveys in 2021, 2022, and 2023 and made 76 captures of 32 individuals, all adults.

### Annual survival and population sizes

The most parsimonious capture-mark-recapture model had capture probability $p$ and annual adult survival $\varphi$ varying by site (Table 2). Models with no movement generally performed better than those with random emigration (Table 2). The top model estimated population sizes ($\hat{N}$) ranging from 13–137 adult frogs per site across years (Figs. 5–7). The highest $\hat{N}$ estimate was from Fohisokina in 2015 (137, 95% CI [120–170]) and the lowest from Soamasaka in 2023 (13, 95% CI [11–22]). Fohisokina showed a decreasing population size during 2015–2023 (Fig. 5), whereas Soamasaka and Ambatofotsy were relatively stable over a shorter period (Figs. 6 and 7). There was strong support for site-varying survival (Table 2), with the estimated annual adult survival more precise for Fohisokina than Soamasaka (Fig. 8). For Ambatofotsy, the survival estimate was too imprecise to be informative because the data spanned only 3 years and only a small number of frogs were captured each year. Overall, the annual survival estimates at Fohisokina and Soamasaka were comparable, although the estimate from Soamasaka was lower than Fohisokina (Fig. 8).

## DISCUSSION

Our results demonstrate *M. cowanii* was still present at no less than ten localities in 2022–2023, but the population size is very small for at least three sites (<50 adults per site). Extrapolating across all known localities, in the worst-case scenario, the total adult

**Table 2 Comparison of robust design capture-mark-recapture models used for estimating demographic parameters of *Mantella cowanii* populations.** Models are sorted by adjusted Akaike Information Criterion (AIC$_c$). $\varphi$ = annual survival. $p$ = capture probability, parameterized the same as $c$, recapture probability, in all models. $\gamma''$ and $\gamma'$ are temporary emigration parameters, either constrained to 0 for no movement or set equal for random emigration. Site is a group-level effect of localities with three or more years of surveys: Ambatofotsy, Fohisokina, and Soamasaka. The covariate *people* is the number of observers on a survey and *effort* the number of people * the survey length in minutes. $K$ = number of parameters. $-2LL = -2 *$ log-likelihood, a measure of model fit. All models include a random effect of individual on capture probability ($p$) to account for individual heterogeneity in detection.

| Model | $K$ | $\Delta$ AIC$_c$ | Weight | $-2LL$ |
|---|---|---|---|---|
| $\varphi(\text{site}), \gamma''(=0), \gamma'(=1), p(\text{site})$ | 7 | 0.00 | 0.51 | 3,147.78 |
| $\varphi(\text{site}), \gamma'' = \gamma', p(\text{site})$ | 8 | 2.05 | 0.18 | 3,147.78 |
| $\varphi(\text{site}), \gamma''(=0), \gamma'(=1), p(\text{year})$ | 9 | 2.61 | 0.14 | 3,146.28 |
| $\varphi(\text{site}), \gamma''(=0), \gamma'(=1), p(\text{people})$ | 6 | 3.89 | 0.07 | 3,153.71 |
| $\varphi(\text{site}), \gamma'' = \gamma', p(\text{year})$ | 10 | 4.67 | 0.05 | 3,146.28 |
| $\varphi(\text{site}), \gamma'' = \gamma', p(\text{people})$ | 7 | 5.93 | 0.03 | 3,153.71 |
| $\varphi(\text{site}), \gamma''(=0), \gamma'(=1), p(\text{effort})$ | 6 | 7.03 | 0.02 | 3,156.85 |
| $\varphi(\text{site}), \gamma'' = \gamma', p(\text{effort})$ | 7 | 9.07 | 0.01 | 3,156.85 |
| $\varphi(.), \gamma''(=0), \gamma'(=1), p(\text{site})$ | 5 | 12.32 | 0.00 | 3,164.17 |
| $\varphi(.), \gamma''(=0), \gamma'(=1), p(\text{year})$ | 7 | 12.76 | 0.00 | 3,160.54 |
| $\varphi(.), \gamma''(=0), \gamma'(=1), p(\text{people})$ | 4 | 14.18 | 0.00 | 3,168.07 |
| $\varphi(.), \gamma'' = \gamma', p(\text{site})$ | 6 | 14.35 | 0.00 | 3,164.17 |
| $\varphi(.), \gamma'' = \gamma', p(\text{year})$ | 8 | 14.81 | 0.00 | 3,160.54 |
| $\varphi(.), \gamma'' = \gamma', p(\text{people})$ | 5 | 16.21 | 0.00 | 3,168.07 |
| $\varphi(.), \gamma''(=0), \gamma'(=1), p(\text{effort})$ | 4 | 17.37 | 0.00 | 3,171.25 |
| $\varphi(.), \gamma'' = \gamma', p(\text{effort})$ | 5 | 19.40 | 0.00 | 3,171.25 |
| $\varphi(.), \gamma''(=0), \gamma'(=1), p(.)$ | 4 | 21.70 | 0.00 | 3,175.58 |

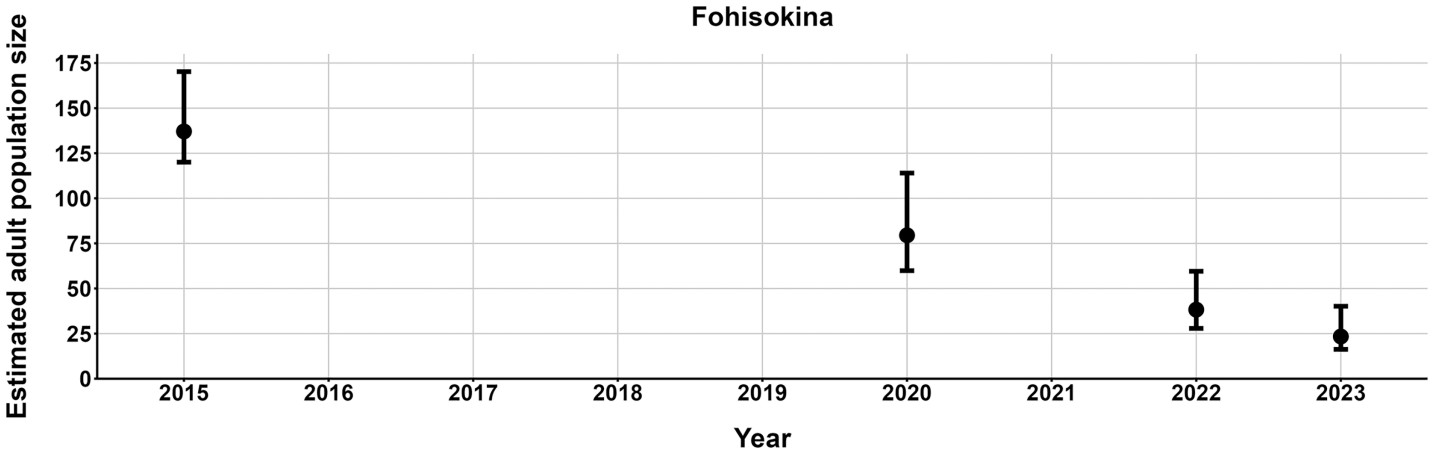

**Figure 5 Changes in the estimated adult population size of *Mantella cowanii* at Fohisokina in the central highlands of Madagascar.** The estimates are from the top model in Table 2. Error bars are 95% confidence intervals around the estimate. Years without points and error bars were not surveyed.

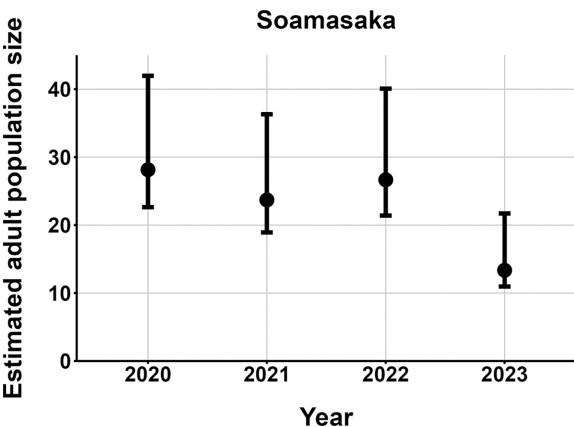

**Figure 6 Changes in the estimated adult population size of *Mantella cowanii* at Soamasaka in the central highlands of Madagascar.** The estimates are from the top model in Table 2. Error bars are 95% confidence intervals around the estimate.

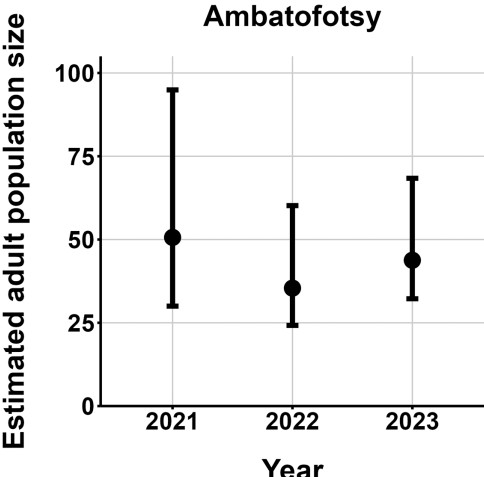

**Figure 7 Changes in the estimated adult population size of *Mantella cowanii* at Ambatofotsy in the central highlands of Madagascar.** The estimates are from the top model in Table 2. Error bars are 95% confidence intervals around the estimate.

population size for the species may number <500 individuals. However, frog populations naturally fluctuate in abundance, and a snapshot over several years can easily lead to erroneous conclusions that populations are declining when they are stable (*Pechmann et al., 1991*; *Blaustein, Wake & Sousa, 1994*; *Meyer, Schmidt & Grossenbacher, 1998*). Such fluctuations are typical of species with fast life histories, where fecundity is high and generation time short, thus, demographic rates tend to vary (*Sæther et al., 2004*, *2013*). Additionally, amphibian populations fluctuate more for pond breeding species and less for terrestrial and stream breeding species (*Green, 2003*). Considering *M. cowanii* is a terrestrial stream-breeding frog with comparatively low reproductive output (20–57 eggs per egg mass; *Tessa et al., 2009*), relatively high annual adult survival, and a maximum lifespan in nature of at least 9 years, we believe our results are not an artifact of stochastic fluctuations in population size.
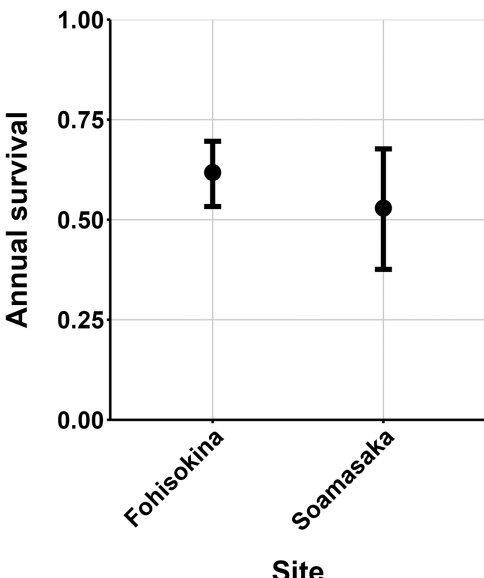

**Figure 8** Estimated adult annual survival of *Mantella cowanii* in two populations. The error bars are 95% confidence intervals around the estimate.

We highlight three possible factors contributing to the >80% population decline at Fohisokina between 2015 and 2023. First, fires burnt much of Fohisokina in November 2020 at the start of their breeding season, just before our survey. Though *M. cowanii* presumably is protected from fire while sheltering in moist rock crevices and caves for much of the year, fire can cause mortality in terrestrial frogs when they are active above ground during the breeding season (*e.g.*, *Humphries & Sisson, 2012*; *Potvin et al., 2017*). Second, according to a European private breeder, in 2017 more than 100 *M. cowanii* were illicitly offered for sale in Germany, and an unknown number were offered again in 2021. As Fohisokina is the most easily accessed *M. cowanii* locality and was historically a collection site for the pet trade (*Rabemananjara et al., 2007b*), the frogs were possibly poached from Fohisokina. Lastly, although no records of chytridiomycosis have been confirmed in Madagascar and all frogs we sampled appeared healthy, the amphibian chytrid fungus *Batrachochytrium dendrobatidis* has been reported from the island, and in 2014 was detected on a single *M. cowanii* individual at Soamasaka ~5 km south of Fohisokina (*Bletz et al., 2015*). Therefore, we cannot rule out disease either.

At Vatolampy, we did not detect *M. cowanii* after 6 days of surveys and suspect the population is extirpated. Until our work, the site had not been surveyed since 2003–2004, when *Andreone et al. (2007)* and *Rabemananjara et al. (2007a)* collected tissue samples and voucher specimens from the population. Similarly, Andraholoma had not been surveyed since 2009 when the site was visited by one of us (C. Randrianantoandro) for one day and five individuals observed. Conversely, we question whether Tsimabeomby, the third site where we did not detect *M. cowanii*, ever supported a population. We believe the locality was possibly published in error by *Rabibisoa (2008)* because Tsimabeomby consists of a wet meadow, is without rocks or running water, is isolated from the next nearest population by >3 km, and the local people we worked with had never encountered

*M. cowanii* there whereas they knew of the other populations. Nonetheless, *M. cowanii* could have been present but undetected during our surveys if the detection probability at Andraholoma, Tsimabeomby, and Vatolampy was lower than elsewhere. Additionally, the naïve estimate of detection probability we used to infer absence did not account for observer, environmental, or temporal factors influencing detection. Still, our team detected *M. cowanii* at other sites on the days we surveyed Andraholoma, Tsimabeomby, and Vatolampy, illustrating local environmental conditions were favorable for detecting *Mantella*. We recommend re-surveying the three sites in the coming years to confirm if the populations are extirpated, especially considering the site-level variation in capture probability we found at sites where *M. cowanii* was present.

Some amphibian species are capable of dispersing long distances to recolonize suitable habitat (*Marsh & Trenham, 2001*; *Fonte, Mayer & Lötters, 2019*), but we do not expect so for *M. cowanii* in Madagascar's degraded highland landscape. There are no studies on the movement of Madagascar's poison frogs (but see *Andreone et al., 2013*), however research on their Neotropical dendrobatid counterparts shows adults rarely move more than a few hundred meters from established territories (*e.g.*, *Ringler, Ursprung & Hödl, 2009*; *Pašukonis et al., 2013*; *Beck et al., 2017*; *Pašukonis, Loretto & Rojas, 2019*). Moreover, in a review of the dispersal ability of amphibians, *Smith & Green (2005)* found nearly half of studied species moved <400 m. Historically, adult frogs may have moved between patches when there was more forest in the highlands. However, we suspect *Mantella* usually passively disperse when tadpoles are flushed between habitat patches during heavy rain. As such, natural recolonization of Vatolampy or Andraholoma is unlikely, especially considering the physiological, movement, and site fidelity constraints amphibians face (*Blaustein, Wake & Sousa, 1994*). At Vatolampy, the next closest locality is Farihimazava, 1.5 km northeast in a different valley, which supports mostly *M. baroni* and *M. baroni* × *M. cowanii* hybrids (*Chiari et al., 2005*; *Andreone et al., 2007*). For Andraholoma, the next closest locality is Andaobatofotsivava >7 km north, though admittedly, the area of the Itremo Massif is poorly explored and there could be additional unrecorded populations between the two.

Previous skeletochronology research estimated the maximum lifespan of *M. cowanii* at 3 years (*Guarino et al., 2008*; *Andreone et al., 2011*), but we identified individuals at least 8 years post-metamorphosis and one individual 9 years. Skeletochronology is known to underestimate the ages of older individuals because skeletal growth rings progressively converge with age, and amphibian bone tissue is prone to reabsorption (*Eden et al., 2007*; *Sinsch, 2015*). Our results show the advantages of using capture-mark-recapture surveys for estimating demographic traits if resources are available. The long lifespan of *M. cowanii* is notable when considered together with their reproductive output and body size. The species is one of the largest in the genus, has the largest egg diameter, and has the lowest number of eggs per mass, exemplifying their slow life history compared to other *Mantella* species (*Tessa et al., 2009*). All of this aligns with our discovery that *M. cowanii* has the longest lifespan for the genus. Life history traits often follow altitudinal clines, with slower traits associated with higher altitudes (*Hille & Cooper, 2015*; *Laiolo & Obeso, 2017*). Indeed, amphibians tend to live longer and have larger body sizes at higher altitudes

(*Morrison & Hero, 2003*; *Andreone et al., 2004*); *M. cowanii* is no exception (*Tessa et al., 2009*). Such patterns also occur within species across altitudinal gradients (*Zhang & Lu, 2012*). Considering the *M. cowanii* from Betafo populations occur ~500 m higher than where we recorded ~8–9-year-old individuals, frogs from Betafo may live even longer. Such slow life history traits also have conservation implications because they are associated with higher extinction risk (*Webb, Brook & Shine, 2002*). Given the relatively long lifespan and low reproductive output of *M. cowanii*, the success of recovery efforts for the species may not be as rapid as in other related amphibian species.

Local people brought our attention to two new *M. cowanii* localities during fieldwork, highlighting the value of community engagement when conducting research on threatened species. To our knowledge, the new localities had not been identified before but were noticed after our initial work conducting surveys together with local communities during 2020–2021. Indeed, people who live in biodiverse rural areas have a unique opportunity to assist in ecological research and monitoring programs (*Schmiedel et al., 2016*). Such opportunities are especially present in Madagascar (*e.g.*, *Dolch et al., 2015*; *Price, Randriamiharisoa & Klinges, 2023*), where most people are subsistence farmers in rural areas and often depend heavily on forest resources. By actively participating and being included in fieldwork, local people recognized the significance of observing *M. cowanii* at new sites, helping inform conservation efforts.

To ensure populations remain extant, we must better identify the causes of declines and the magnitude of threats. Screening populations for *Bd* and monitoring for illicit *M. cowanii* in the pet trade are essential actions, while disentangling the complex threat of habitat loss presents additional challenges. The number of trees remaining at sites varies from intact closed-canopy forest to rocky landscapes almost entirely devoid of trees, so the degree to which deforestation is a threat may depend on additional habitat characteristics. *Newton-Youens (2017)* identified rock caves and refuges as essential habitat features, and speculated they might be used for breeding, though so far, no eggs, tadpoles, or newly metamorphosed individuals have been found in nature. Studies on the microhabitat preferences and activity levels of *M. cowanii*, like those carried out by *Edwards et al. (2019)* and *Edwards, Bungard & Griffiths (2022)* for *M. aurantiaca* and *Rasoarimanana, Edmonds & Marquis (2024)* for *M. baroni*, would further help identify the most critical habitat features to protect and the best time to survey sites. Likewise, better information about habitat requirements could be used to locate new sites with unprotected populations we do not know about. All known *M. cowanii* populations are centered around four isolated sites with likely past connectivity when the highlands were an intact forest-grassland mosaic (*Bond, Silander & Ratsirarson, 2023*). Fieldwork in the remote areas between the four population centers could uncover additional isolated *M. cowanii* populations, but time is running out.

## CONCLUSIONS

We set out to verify the presence and estimate key demographic traits for one of Madagascar's most threatened frog species and found three historical localities may be extirpated while other populations are extremely small. Unfortunately, our results are

not unique to Madagascar but represent a global trend in amphibian populations (*Stuart et al., 2004*; *Grant et al., 2016*). Amphibians are at the forefront of the extinction crisis, and population monitoring is essential to measure responses to conservation actions and detect declines before recovery is impossible. We used capture-mark-recapture methods to estimate abundance at three localities (Ambatofotsy, Soamasaka, and Fohisokina; Fig. 3), but less costly approaches relying on presence-absence data are likely suitable for monitoring *M. cowanii* across its entire range (*Joseph et al., 2006*; *Jones, 2011*). When enacted with local people as part of a broader program, monitoring can galvanize conservation efforts by adding value to a threatened species and instilling pride in local communities (*Andrianandrasana et al., 2005*; *Danielsen, Burgess & Balmford, 2005*). Given that adequate information on a species basic ecology and life history is essential to addressing the causes of decline, we recommend further research run concurrently with conservation efforts and focus on determining the relative impact of disease, illegal trade, and habitat loss. We also recommend reassessing the IUCN Red List status of *M. cowanii*. The species was last assessed in 2014 as Endangered, but it may qualify for the Critically Endangered status based on our estimates of population sizes and trends.

## ACKNOWLEDGEMENTS

We are grateful to Chloe Helsey who assisted with fieldwork in 2015, to L'Homme et l'Environnement and V.O.I. FOMISAME for collaborative efforts at Antoetra, and to Tiana Randriamboavonjy of the Kew Madagascar Conservation Centre for logistical support at Itremo. Special thanks go to Association Mitsinjo members Frederic R. Razafimahefa, Georges Ramarolahy, Samina S. Sam Edmonds, Edupsie Tsimialomanana, and J.E.A. Fanirihasimbolatiana for providing indispensable field assistance. We thank R. Griffiths, E. Larson, and R. Schooley for offering feedback on an early draft of the manuscript.

### Funding

This work was supported by the Amphibian Survival Alliance, Mohamed bin Zayed Species Conservation Fund, Parc Zoologique de Paris, American Frog Day, Sean Betti and Infinite Networks, Parco Natura Viva, Foundazione ARCA, and Chester Zoo. The Illinois State Toll Highway Authority funded work on the analysis and publication charges. Travel expenses for Devin Edmonds were provided by the University of Illinois Urbana-Champaign ACES Office of International Programs and ACES Education Abroad. The Portuguese National Funds through FCT (Fundação para a Ciência e a Tecnologia) supported the research contract to Angelica Crottini [2020.00823.CEECIND/CP1601/CT0003]. The funders had no role in study design, data collection and analysis, decision to publish, or preparation of the manuscript.

## Grant Disclosures

The following grant information was disclosed by the authors:

Amphibian Survival Alliance.
Mohamed bin Zayed Species Conservation Fund.
Parc Zoologique de Paris.
American Frog Day.
Sean Betti and Infinite Networks.
Parco Natura Viva.
Foundazione ARCA.
Chester Zoo.
Illinois State Toll Highway Authority.
UIUC ACES Office of International Programs.
FCT (Fundação para a Ciência e a Tecnologia): 2020.00823.CEECIND/CP1601/CT0003.

## Competing Interests

The authors declare that they have no competing interests.

## Author Contributions

- Devin Edmonds conceived and designed the experiments, performed the experiments, analyzed the data, prepared figures and/or tables, authored or reviewed drafts of the article, and approved the final draft.
- Raphali Rodlis Andriantsimanarilafy conceived and designed the experiments, performed the experiments, authored or reviewed drafts of the article, and approved the final draft.
- Angelica Crottini conceived and designed the experiments, authored or reviewed drafts of the article, and approved the final draft.
- Michael J. Dreslik conceived and designed the experiments, authored or reviewed drafts of the article, and approved the final draft.
- Jade Newton-Youens conceived and designed the experiments, performed the experiments, authored or reviewed drafts of the article, and approved the final draft.
- Andoniana Ramahefason performed the experiments, authored or reviewed drafts of the article, and approved the final draft.
- Christian Joseph Randrianantoandro conceived and designed the experiments, authored or reviewed drafts of the article, and approved the final draft.
- Franco Andreone conceived and designed the experiments, authored or reviewed drafts of the article, and approved the final draft.

## Animal Ethics

The following information was supplied relating to ethical approvals (*i.e.*, approving body and any reference numbers):

The research was approved by the University of Illinois at Urbana-Champaign Institutional Animal Care and Use Committee (Protocol #21180).

## Field Study Permissions

The following information was supplied relating to field study approvals (*i.e.*, approving body and any reference numbers):

The Ministère de l'Environnement et du Développement Durable of Madagascar approved the study (N°173/20/MEDD/SG/DGGGE/DAPRNE/SCBE.Re, N°439/21/MEDD/SG/DGGGE/DAPRNE/SCBE.Re, N°173/22/MEDD/SG/DGGGE/DAPRNE/SCBE.Re).

## Data Availability

The data and code are available at the Illinois Data Bank:

Edmonds D, Andriantsimanarilafy R, Crottini A, Dreslik M, Newton-Youens J, Ramahefason A, Randrianantoandro CJ, Andreone F. 2024. Data and code for estimating population sizes, annual survival, and inferring absence of the frog *Mantella cowanii*. University of Illinois at Urbana-Champaign. https://doi.org/10.13012/B2IDB-0681943_V1.

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
