# Peer review of "Small population size and possible extirpation of the threatened Malagasy poison frog Mantella cowanii"

_PeerJ, doi:10.7717/peerj.17947_

## Round 0.1 · original submission · Major Revisions

Thank you for your submission. I have received two very detailed reviews of your manuscript. Both reviewers agreed that your manuscript is well written and seems to be based on a solid mark-recapture design. However, there were a number of points raised by the reviewers that must be addressed before this submission can be further considered. Most of the identified issues are points in need of further clarification where additional text in the manuscript is warranted to better communicate what was done. I do not think that any critical flaws were identified, but the quantity of comments, range of topics, and depth of issues to be remedied rose to the level of major revision. I look forward to seeing your revised manuscript.

·

Basic reporting

I would like some more information about the prior knowledge of M. cowanii populations. I can understand that the authors are intentionally using more vague language to protect the extant populations of frogs, but it stylistically doesn’t match the rest of what is overall a well-written paper.
• More specifically, I want to know if there is a citable source that refers to the “one or two scientific expeditions” that would tell me a) who led these expeditions or b) when “decades ago” was (were these in the 2000s, 1980s?, or the 1920s? earlier?).
• Similarly, the authors introduce the McAP program as an initiative to map out conservation plans for the species, but what does a “blueprint for its conservation” entail? How do the goals of the paper reflect and advance the goals/missions of McAP? The authors cite the Rabibisoa 2008 conservation plan, but a brief mention in text would help readers that aren’t familiar with McAP and better situate the author’s work within larger initiatives. I think the knowledge gap addressed by the paper is apparent, but situating the knowledge gap more clearly within the context of prior literature would advance the author’s arguments.

I think for the results section to better reflect the discussion, there needs to be some connecting piece between the annual survival and population estimates that are disclosed in the results and the 80% population decline at Fohisokina that is a focal point in the discussion (line 259-260). I agree with the suggestions from prior reviewers that Figure 5 clarifies the results, but I think more of the take-home messages for this figure should be written in the text.

Minor comments:
Line 59: The in-text citation should be Collins, 2010; Grant, Miller & Muthus, 2020). Campbell is actually Evan’s middle name. This change should also be made where you cite the same paper on line 83.
Line 87: “All” technically refers to the singular species of Mantella (M. laevigata), but I think you mean all members of the genus.
Line 90: “While several species” technically refers to “Neotropical dendrobatids”, and I think it gets a little confusing talking about multiple families/genera of frogs without explicitly specifying.

Experimental design

Line 144: I think there needs to be some kind of qualifier around the words “active” and especially around “detectable”. The way the sentence currently reads implies that M. cowanii estivates from late Jan through mid November, and the species is not active or detectable during most of the year.

I would like some more context for the author’s decision to conduct all but the 2015 Fohisokina surveys during the early morning. Are M. cowanii more likely to be surface active/visible during this time? Do the frogs call more during the morning than at other times of the day? If there are differences in activity for the species, were these differences controlled for in the authors’ detection models? Similarly, I would like the authors to confirm if the dusk surveys occurred on the same day as dawn surveys and if so, are the authors counting the “number of days” separately for dawn and dusk surveys when they report survey effort (Table 1).

Did the authors count detections if they only heard the frog? If so, is there a sex or stage class bias in calling behavior (ie. Only adult males call) that may influence detection over using visual surveys alone? I would be less concerned about this bias if the authors only use the calls to locate animals and then count detections when they had individuals in-hand, but I think a distinction between the field protocols for the occupancy surveys at the multiple sites and the CMR surveys at the 3 sites would be helpful.

Line 203-204: Based on the timing of the authors’ surveys, what represents an independent survey? Is this simply when the population is not assumed to be closed? What is the time period between the 2 primary periods (“up to two times” on line 136-137 ) and the 1-6 days? Do these represent the years denoted in Table 1? If so, I think referencing Table 1 would help clarify the authors’ variation in survey effort. Are these 1-6 days consecutive? In table 1, does the “Days” column refer to the number of primary periods (with varying numbers of secondary periods that aren’t disclosed) or is “Days” the number of surveys (pooled across primary periods)?

Given that the authors surveyed the 13 “known” sites, how did they discover the 2 new sites? Again, I can understand that they would want to limit the information that is disclosed about a lesser-known and population, but I would like some information about how they ended up surveying at these new sites in addition to the 13 originally planned sites. Are there other locations that they surveyed where they failed to detect animals? Those absence datapoints are also useful when thinking about species distribution and habitat use.

I want to know more about how the authors chose the locations for surveys. To whom are these 13 localities known? The scientific community? Members within McAP? I think some citation, if available, that refers to prior work on the distribution of the species would be helpful. I think this is a language issue, but without specifying, I think it opens up some philosophical questions about local or traditional ecological knowledge versus conventional or “technical” science, which the authors hint at in the discussion about including local communities in conservation solutions.

Minor comments

Line 188-189: Depending on how remote the study sites are, this may not have been an option, but are there not larger satellites or weather station datasets that you could utilize (PRISM, some Malagasy equivalent to NOAA)? I understand that this also complicates things with getting regional climate versus microclimates, but there are other ways to collect environmental data other than while you’re actively surveying. I did a quick google search and found this resource that could potentially be helpful. https://climateknowledgeportal.worldbank.org/country/madagascar/climate-data-historical.

I recognize that there is no perfect way to individually “mark” animals in capture mark recapture studies. I also recognize the need to use minimally invasive techniques given the conservation concern of the species, but you contradict yourself from line 160-161 and 162-165. Image recognition can be useful for smaller datasets, but in my experience, they begin to be less accurate after a certain number of animals are introduced into the system (we worked with spotted salamanders, who also theoretically have unique dorsal patterns). I think you need to define what you mean by “more accurate (line 160) and recognize this may not be the case in all situations. I would just frame your choice for how you mark animals within the context of your own research goals/system constraints, rather than some methods inherently being better than others.

I would like the authors to either expand on why 3 years is the minimum for estimating annual survival or provide a citation. I could understand if they were measuring mean annual survival, which would require at least 2 intervals to estimate survival (and have some variation), but I don’t understand why one couldn’t theoretically estimate survival for a singular interval.

I needed to do a refresher to remind myself what the difference between y’ and y’’ are. I was also confused by the assumptions about population closure across the entire study period given the parameterization of y’ and y” (Line 181-183). I think the authors make a lot of assumptions about their reader’s prior knowledge of robust design capture mark recapture models. I would like some brief descriptor about model structure/closure assumptions, especially given my questions about timing between surveys.

Lines 236-238: I am concerned by the author’s inference that the difference in adult survival estimate precision is driven by the longer study duration at Fohisokina. Although the number of years is different (8 and Fohisokina versus 4 at Soamaska), because there is no data from 2015 to 2020, the number of primary periods is the same between Fohisokina and Soamaska (n=4). Therefore, the number of intervals that would go into the annual survival estimate is the same. Along that same line, I would like the authors to confirm that they accounted for the time in between surveying seasons in their survival models? (i.e. the probability that an animal survives between time step 1 (2015) and time step 2 (2020) is not the same as the probability of it surviving from 2020 to 2021).

Even with the revisions and clarifications from previous reviewers, I am still concerned about the conclusions that the authors draw from their analyses, especially given the site-level variation in survival and detection probability.
• It is interesting to me that despite the wide variation in sampling effort, those factors were not included in the most supported model. I commend the authors for including these controls in their analyses, but I think this further reiterates the need to include some kind of caveat about designating sites as extirpated given the limited sampling effort and spatial heterogeneity in detection probability.
• Given the relatively higher survey effort at Vatalampy, and repeated “absence” records for 2022 and 2023, I feel more confident that the frogs at Vatolampy are extirpated. However, the authors did find frogs at Bekaraka with 3 surveys but are calling the population extant 2 years later with only 2 surveys. Based on this table, it is unclear to me if the frogs were detected in the 1st 2 visits in 2021 or if frogs were only detected in the 3rd visit in 2021. Additionally, the authors are classifiying the Tsimabeomby population as extirpated after only 1 survey. I would just be really cautious about calling this site extirpated, given the variation in detection probability at each site (the best model has survival AND detection varying by site).
• Line 356-357: “several” I think oversells the number of extirpated sites. I think this is a precision of language issue more than anything, but especially given my concerns about the Bekaraka and Tsimabeomby sites, I would prefer the authors replace “several” with “three”.

For Figure 5, I think it is it misleading to have the years spaced equally across the x axis at Fohisokina. Visually, the gaps in data (2015-2020 and 2021) are not readily apparent, and it makes the decline in population abundance seem more rapid than it really is, given that we don’t know how fast the population declined in the years that weren’t surveyed, and the time across the x axis is longer in 2015-2020 than in 2022 to 2023.

Validity of the findings

Lines 209-210 (and then discussed on (line 311 and 312): I would like more information about these previously unrecorded populations. Based on the experimental design/field methods, it is unclear to me how you decided to survey at those locations? When did you find them (2015, or 2023)? I understand not wanting to disclose exact locations, but broad areas relative to the known locations would help identify possible gaps in the range or range shifts where there may be other populations that aren’t known to science.

Line 365-369: I think the point about enacting monitoring plans in conjunction with broader programs that collaborate with local people is great. I think this introduces a new idea that should really be in the discussion (and then reiterated in the conclusion). If the authors choose to discuss this point (which again I commend them for), I would like to see this practice highlighted in their methods. Which broader program(s) are the authors working with (see my point about further connecting your work to McAP), and how are the authors supporting the capacity building of local people so that they are empowered to take ownership of local biodiversity challenges? How does the authors’ work “add value to a threatened species and “instill[ing] pride in local communities” (line 365-367) in a way that equitably includes local communities and fosters empowering collaborations (i.e. avoiding “helicopter science”) without further promoting illicit pet trade?

Minor comments
I think the more “obvious actions” downplays the complexities of understanding Bd dynamics in a complex ecosystem and the logistical constraints that go into screening and sampling for Bd. I think my point is further reiterated by your response to prior reviewers about complications for monitoring and screening for Bd. Similarly, I think monitoring for illicit M. cowanii pet trade is really complicated and equally confronts logistical constraints, and legal, cultural, sociopolitical challenges…as you pointed out with the “unknown number” (line 266) of frogs “offered in 2021”. I understand that this is intended to set up the uncertainty in habitat loss in contrast to other drivers for decline, but I think there are ways to do this that don’t suggest screening and monitoring are “obvious actions” that “ensure populations remain extant”.

Line 368: The “as we have shown” suggests that you explored the causes of declines in your methods/analyses. I think “As we have discussed” would be a more appropriate segue way or “Given that adequate information on a species basic ecology and life history is essential to addressing the causes of decline, we recommend further research…determining the impact of disease, illegal trade, and habitat loss.”

Additional comments

The authors use capture mark recapture analyses to estimate population vital rates (N-hat, and survival). They use data following a robust design structure to estimate abundance and survival for 3 populations of M. cowanii, a threatened Malagasy frog. Additionally, they report presence/absence data for 10 other locations that were previously known to support M. cowanii populations. In this manuscript, the authors discuss the conservation status of M. cowanii, prior work and ongoing challenges for population management. Generally, I think the authors do a good job of laying out the threats to M. cowanii populations and situating them within the larger trends in amphibian biodiversity loss. I think with some further clarification in field method protocols and model construction, this manuscript will be greatly improved. While the manuscript is clear, concise, and generally well-written, I found some points that I think would benefit from further clarification or the addition of additional nuance or context.

I include comments in each of the prescribed headers (as requested by the journal) which are generally bigger talking points or concerns related to each section. I also include the subheader “Minor comments” for feedback that I interpret more as stylistic edits or minor revisions. Highlighted text in the tracked changes word document corresponds to the comments outlined in this document, but I tried to include line numbers when relevant.

·

Basic reporting

It is an interesting well written and structured article on a threatened species with restricted and fragmented distribution range. This is the most up dated information on the arlequin mantella with respect to distribution ranges, demographic traits, survival, and population size.
However, there are few typos to correct, clarifications to provide, and suggestions to consider to better appreciate the relevance and importance of this manuscript, with regard to its valorization in the field of management and conservation of this species. All of these cited points are inserted in the word version of the manuscript attached in this report.

Experimental design

Appropriate. Just few minor comments already inserted in the word version.

Validity of the findings

Reliable and relevant findings, however, one must be careful in the interpretation of the data to avaoid any hasty conclusions.

Additional comments

This is the most updated information on different aspect of the endangered species Mantella cowanii. However, as the authors already underline many more fieldworks are necessary to fill in the gap. The contribution of the present article in the implementation of the action plan for the conservation of this species is non negligible. It is important to focus more attention of the ecological requirement of this target species.

---

## Round 0.2 · Minor Revisions

I appreciate the author's willingness to make suggested changes and revise their manuscript accordingly in the first round. I think that the manuscript is much improved for such efforts. I have now received comments from one of the original reviewers, who has made several, mostly minor, suggestions for the authors to consider during their revisions and a few clarifying comments. Once these been adequately addressed, I would be happy to accept this paper at PeerJ.

·

Basic reporting

Line 336: I was curious about the Smith and Green 2005 review for my own reading, but I was unable to find it in the authors' References list. I did not go through and check all of their in text citations/listed references, but if it truly isn’t there, please make sure your in text references are all included in your list!

Lines 160-163: I thank the authors for adding context for their choices in the experimental design. While I think this context is valuable, it might be better in the introduction (with a brief reference to “given the activity patterns of the species…”, more specifically in the paragraph describing the ecology and distribution of M. cowanii. I don’t feel strongly about this stylistic suggestion, but I mention it only because the content before and after reads more like a traditional “methods” section while these lines feel more like introductory context.

Line 250: Similar to the point for lines 160-163 about connecting the introduction + methods, I didn’t know that M. cowanii hybridizes with M. baroni (again, knowing nothing about your study system). Are there concerns for the genetic diversity/differentiation between species? Are there concerns with competition between the parent species? I think a quick note in the introduction to a) contextualize M. cowanii conservation within this other component and b) reiterate the author’s choice to only use calls to locate individuals (thank you for clarifying from previous versions). Similarly, are hybrids visually distinct from parent species? How many hybrids did they find (I know this isn’t the main focus, but it could be helpful to get at how big of an issue the hybridization is from a conservation perspective).

Line 37-38: This is another stylistic/philosophical point than a methods issue. However, I think the authors need to be careful about saying *they* discovered two new populations if local community members were aware of M. cowanii in these locations. We “describe two additional populations” or we “discovered two new populations to the scientific literature”.

Line 412-422: Adding a comma between “Endangered” and “but” to make it a compound sentence.

Experimental design

no comment

Validity of the findings

Line 44: The authors’ results show the survival rate and a significant increase in the known life span of M. cowanii, and they argue for the value of capture-mark-recapture surveys. This is all great, but I think these results alone paint an incomplete picture of where M. cowanii is on the pace of life spectrum. I think the discussion (Lines 349-371) fills in a lot of the pace of life factors, but it leans on a lot of findings outside this manuscript. I think something like “Our results reflect prior work that M. cowanii is characterized by a slower life history pace relative to other Mantella…”

Additional comments

The authors made significant changes since I reviewed the manuscript in earlier rounds of revisions. The additional information on the Amphibian Specialist Group of Madagascar and Conservation International’s efforts (including McAP) was helpful to contextualize the authors’ work within larger conservation + monitoring efforts. They also added ecological and methodological context to better support their experimental design and analysis decisions, which were my main concerns from the previous review. All of my comments at this point are minor changes that I believe will help polish the manuscript before publication. I'm including an annotated version of the manuscript with highlighted sections corresponding to my comments.

---

## Round 0.3 · accepted · Accept

Thank you for addressing all of the remaining comments. I am happy to accept this article at PeerJ. Thank you for your submission and congratulations!